# Epidermal Growth Factor Stimulates Fatty Acid Synthesis Mainly via PLC-γ1/Akt Signaling Pathway in Dairy Goat Mammary Epithelial Cells

**DOI:** 10.3390/ani10060930

**Published:** 2020-05-28

**Authors:** Jiangtao Huang, Bangguo Dai, Hexuan Qu, Yuling Zhong, Yue Ma, Jun Luo, Darryl Hadsell, Huaiping Shi

**Affiliations:** 1Shaanxi Key Laboratory of Molecular Biology for Agriculture, College of Animal Science and Technology, Northwest A&F University, Yangling 712100, China; jiangtaoh@nwafu.edu.cn (J.H.); Dai_bangguo@163.com (B.D.); quhexuan1995@163.com (H.Q.); zhongyuling3424@163.com (Y.Z.); mayue152@163.com (Y.M.); luojun@nwsuaf.edu.cn (J.L.); 2Pediatrics-Nutrition Childrens Nutritions Research Center, Department of Pediatrics, Baylor College of Medicine, Houston, TX 77030, USA; dhadsell@bcm.edu

**Keywords:** EGF, EGFR, fatty acid, Akt, GMEC

## Abstract

**Simple Summary:**

Goat milk contains an abundance of fatty acids which are benefit to human health. Epidermal growth factor (EGF) is a small peptide which could positively regulate the growth, development and differentiation of the mammary gland during lactation. However, little information is available about EGF in regulating lipid metabolism in the mammary gland. This study investigated the effects of EGF on the triglyceride (TG) synthesis, lipogenic genes expression and the downstream signal protein levels in goat mammary epithelial cells (GMECs). Our findings indicated EGF might be beneficial to improve milk fat synthesis of dairy goats.

**Abstract:**

EGF acts as a ligand of the EGF receptor (EGFR) to activate the EGFR-mediated signaling pathways and is involved in the regulation of cell physiology. However, the roles of EGFR mediated signaling pathways in the regulation of lipid metabolism in goat mammary epithelial cells (GMECs) are poorly understood. To evaluate the impact of EGF on GMECs, the triglyceride (TG) content and lipid droplet were detected, using TG assay and immunofluorescence. Further, expression of lipogenic genes, the protein kinase B (Akt), phospholipase C-γ1 (PLC-γ1) and extracellular signal-regulated kinases (ERK)1/2 signaling pathways were measured by real-time polymerase chain reaction and Western blot, respectively. The results showed that the mRNA expression of *EGFR* gene was significantly upregulated in lactating goat mammary gland tissues compared to non-lactation period (*p* < 0.05). TG contents in EGF-treated GMECs were significantly increased (*p* < 0.05), and an increase of lipid droplets was also detected. In vitro studies demonstrated that the mRNA levels of lipogenesis-related *FASN*, *ACC*, *SCD1*, *LXRa*, *LXRb* and *SP1* genes were positively correlated to the mRNA level of EGFR gene shown by gene overexpression and silencing (*p* < 0.05). The phosphorylations of Akt, ERK1/2 and PLC-γ1 in GMECs were greatly upregulated in the presence of EGF, and specific inhibitors were capable of blocking the phosphorylation of Akt, ERK1/2 and PLC-γ1. Compared with EGF-treated GMECs, the mRNA levels of *FASN*, *ACC* and *SCD1* were significantly decreased in GMECs co-treated with PLC-γ1 and Akt inhibitor and EGF (*p* < 0.05), and TG content was also dropped significantly. These observations implied that EGFR plays an important role in regulating de novo fatty acid synthesis in GMECs, mainly mediated by Akt and PLC-γ1 signaling pathways.

## 1. Introduction

Goat milk is known as a nutritious food with ahigh content of short- and medium-chain fatty acids [1]. These fatty acids are synthesized in goat mammary epithelial cells. Then, triglyceride (TG) droplets are formed and transported out of the cell, where the fatty acids become a component of goat milk. Many reports demonstrated that TG synthesis in GMECs is a very complex process, in which many genes related to fatty acids were involved [2,3,4]. Different studies have shown that epidermal growth factor (EGF) and its receptors can stimulate the expression of fatty acid synthase (*FASN*) and other enzymes responsible for endogenous synthesis of fatty acid; however, it is still not clear how they regulate those genes. Specifically, there are no reports defining the molecular mechanism through which EGF regulates TG synthesis.

EGF is a ligand of EGF receptor (EGFR/erbB-1), and its function is achieved through epidermal growth factor receptor (EGFR)-mediated signaling pathways [5]. EGFR is a member of the receptor tyrosine kinase (RTK) family and undergoes autophosphorylation in C-terminal tyrosine residues after binding to EGF [6,7]. Phosphorylated EGFR induces extracellular signals into cells and activates multiple signaling pathways, such as the MEK/ERK1/2 pathway, PI3K/Akt pathway and PLC-γ1 pathway [8,9,10]. The activation of these pathways, in turn, alters the expressions of target genes, along with protein abundance and activity, to produce the cellular physiological responses. Previous research revealed that EGF increased the density of lipid droplets, which depends on EGFR expression and activation, as well as the individual cellular capacity for lipid synthesis [11,12]. Activation of Akt induces the synthesis of sterol regulatory element binding protein (SREBP) protein, and the latter promotes a series of gene expressions in fatty acid synthesis, including *FASN*, acetyl-CoA carboxylase alpha (*ACACA*) and stearoyl-CoA desaturase 1 (*SCD1*) [13]. Elevated FASN expression by the EGFR/ERK pathway induces cell proliferation [14]. Tyrosine-phosphorylated PLC-γ1 enriched in lipid rafts responds to EGFR activation by generating the lipid second messenger diacylglycerol (DAG) [15,16]. 

However, to our knowledge, the molecular mechanisms regarding EGF-mediated signaling pathways in the regulation of lipid metabolism in GMECs remain unclear. The present study aimed to investigate the roles of EGFR-mediated signaling pathways involved in the lipid metabolism of GMECs. Hence, we focused not only on the analysis of EGFR-mediated signaling pathways in GMECs, but also particularly on the lipogenesis molecular events related to EGFR.

## 2. Materials and Methods 

### 2.1. Chemicals 

AG1478 (EGFR inhibitor), MK2206 (Akt inhibitor), U0126 (MEK inhibitor) and U73122 (PLC-y1 inhibitor) were purchased from Selleck Chemicals (Houston, TX, USA). EGF and TRIzol reagent were purchased from Invitrogen Corp. (Carlsbad, CA, USA). Prolactin was purchased from Sigma-Aldrich (St. Louis, MO, USA). Lipofectamine™ 2000 and RNAiMAX were purchased from Thermo Fisher Scientific, Inc. (Waltham, MA, USA). Unless otherwise specified, all of the chemicals were purchased from China.

### 2.2. Collection of Goat Mammary Gland Tissue 

All procedures in our animal study were approved by the Animal Care and Use Committee of the Northwest A&F University (Yangling, China) (permit number: 15-516, data:2015-9-13). Mammary gland tissues were obtained from four healthy herds of Xinong Saanen goats (average of about 57 kg) at different lactation stages, including early lactation, peak lactation, late lactation (20, 60 and 270 days after parturition, respectively) and dry period (60 days prior to parturition; dry-off on day 10). Percutaneous biopsies were performed from the left or right udder, as described previously, and the GMECs were isolated and purified according to our previously described protocol [17,18]. Six biological samples per period were frozen immediately in liquid nitrogen, after washing with diethylpyrocarbonate-treated PBS until RNA extraction, as previously described [19]. 

### 2.3. Phosphorylation Site Analysis of Goat EGFR

Previously, the cDNA sequence of goat *EGFR* gene was cloned in our lab. The protein sequence was deduced from the cDNA sequence of goat EGFR, using DNAMAN software (http://www.lynnon.com/), and the phosphorylation sites of protein were predicted by using NetPhos 3.1 Server (http://www.cbs.dtu.dk/services/NetPhos/, Copenhagen, Denmark). 

### 2.4. Small Interference RNA Design

The small interfering RNA (siRNA) sequences for the coding region of EGFR were chosen by algorithms provided by Invitrogen (http://rnaidesigner.thermofisher.com/rnaiexpress/, Carlsbad, CA, USA). The siRNA sequences for interference are listed in Appendix A.

### 2.5. Cell Culture and Transfection 

Throughout the experiments, GMECs were cultured at 37 °C, 5% CO_2_, and the basal medium (DMEM/F12 with 5 mg/L insulin, 5 mg/L hydrocortisone, 10 kU/L penicillin/streptomycin) supplemented with 10 ng/mL EGF and 10% fetal bovine serum was changed every 24 h. Cells were grown in basal medium for 24 h and infected with GFP (Control) or EGFR adenoviruses (saved in our lab) at 100 multiplicity of infection for 6 h in FBS-free medium (DMEM/F12 with 0.1% (w/v) bovine serum albumin (BSA) and 2 ug/mL prolactin (Sigma-Aldrich, Inc., St. Louis, MO, USA) ), after which cells were washed and replaced with FBS-free medium. Experiments were performed 48 h after infection. Cells were transfected with 100 nm siRNA against EGFR or control siRNA, using RNAiMAX according to the manufacturer’s protocol. Six hours after transfection, GMECs were replaced with FBS-free medium, harvested after 48 h of incubation and lysed, to prepare the protein or RNA samples for detecting EGFR protein and mRNA levels of genes related to lipid metabolism.

### 2.6. Cell Treatment

GMECs were grown in basal medium, until ~80% confluence, before applying treatments. GMECs were replaced with FBS-free medium for 16 h prior to treatment. Then, the cells were stimulated with 50 ng/mL EGF for 36 h, to detect TG content in GMECs. In some experiments, kinase inhibitors, MK2206 (500 nM), U0126 (10 µM) or U73122 (20 µM) were added respectively or together for 60 min before EGF or DMEM/F12 (control), in order to determine whether a particular signaling pathway was involved in the lipid metabolism response. After 24 h of incubation, GMECs were collected and lysed, in order to detect the mRNA levels of genes related with lipid metabolism. In addition, GMECs co-treated with MK2206, U73122 and EGF for 36 h were collected, to analyze the content of intracellular TG. 

To determine whether ERK1/2, Akt or PLC-γ1 serves as a mediator, MK2206 (500 nM) or U0126 (10 µM) or U73122 (20 µM) was added to the FBS-free medium 60 min before the EGF stimulation. After 60 or 15 min incubation, GMECs were collected to measure the phosphorylation level of Akt, ERK1/2 or PLC-γ1. 

GMECs were treated with 0, 1, 5, and 10 µM AG1478 and EGF (50 ng/mL) for 36 h. After that, the lipid droplets in GMECs were detected, using BODIPY^®^ 493/503 assay (Carlsbad, CA, USA) [20]. GMECs stained with BODIPY 493/503 were imaged, using a Leica TCS SP5 spectral scanning confocal microscope system, according to the method of Yang et al. [20]. 

### 2.7. Total RNA Extraction and Real-Time Quantitative PCR

Total RNAs from goat mammary gland tissue or cells were extracted by using TRIzol reagent, and 1 ug of total RNA was used to synthesize cDNA with PrimeScript RT Reagent Kit with a gDNA Eraser (RR047a, Takara, Japan). Real-time quantitative PCR (RT-PCR) was performed according to the manufacturer’s instructions, using SYBR green (SYBR Premix Ex TaqII, Perfact Real Time; Takara Bio Inc., Shiga, Japan). The *UXT* gene was used as a housekeeping gene [21]. RT-qPCR reactions were performed in a Bio-Rad CFX96 TouchTM system (Bio-Red Laboratories, Inc., Hercules, CA, USA). Primers for RT-PCR are shown in Appendix A. Data were analyzed by using the related quantification (2^−∆∆Ct^) method.

### 2.8. Western Blot Analysis

Cells in experiments (adenoviruses and inhibitors) were collected and lysed for total protein, according to the methods of Zhang et al [19]. Western blots were conducted according to the methods of Xu et al. and Gou et al. [22,23]. In brief, 30 ug of total protein samples from three biological replicates was separated by electrophoresis on 7.5–10% sodium dodecyl sulfatepolyacrylamide gels and then transferred onto nitrocellulose paper and incubated with antibodies against β-actin (CW0096, CW Biotech; 1:1000), EGFR (AF1330, Beyotime; 1:1000), pY1068-EGFR (ab40815, Abcam; 1:1000), pS473-Akt (4060, CST; 1:1000), pT202/Y204-ERK1/2 (4370, CST; 1:1000) and pS1248-PLC-γ1 (4510, CST; 1:1000). Eventually primary antibodies were detected with HRP-conjugated secondary antibodies, followed by light detection with chemiluminescent ECL Western blot detection system (Bio-Rad Laboratories, Inc., Hercules, CA, USA). The densities of bands were analyzed by ImageJ (http://imagej.nih.gov/ij/), and relative protein abundance was normalized to β-actin.

### 2.9. Cellular TG Analysis

Intracellular TG was assayed, using a TG assay kit (Applygen Technologies Inc., Beijing, China), according to the manufacturer’s instructions. The optical density (OD) was detected at 550 nm in a Microplate Reader. Data were normalized with protein concentrations measured with a Pierce BCA Assay Kit (Thermo Fisher Scientific, Waltham, MA, USA).

### 2.10. Statistical Analysis

All experiments included at least three biological replicates. The results were expressed as mean ± SD. Statistical significance was analyzed by SPSS software (version 19, SPSS Inc., Chicago, IL, USA), using Student’s *t*-test or one-way (Analysis of Variance) ANOVA with Tukey test. Significant differences were declared at *p* < 0.05.

## 3. Results

### 3.1. mRNA Abundance of EGFR in Goat Mammary Gland Tissues and Lipid Content in GMECs Treated with EGF

The mRNA levels of the *EGFR* gene of goat mammary gland tissues in all lactation periods were higher than the levels in tissue of the dry period (*p* < 0.05, Figure 1A). Moreover, the expression level of EGFR was ~35-fold higher in early lactation and ~30-fold higher in late lactation, compared to the dry period (*p* < 0.05, Figure 1A). TG content in EGF-treated GMECs was significantly increased compared to EGF-untreated GMECs (*p* < 0.05, Figure 1B). Furthermore, it was found that the fluorescence intensity in EGF-treated GMECs was increased (Figure 1C). However, when GMECs were co-treated with AG1478 and EGF, the fluorescence intensity was significantly decreased compared to EGF-treated GMECs (Figure 1C). In fact, the total fluorescence intensity dropped steadily with the increase of AG1478 concentration (Figure 1C). 

### 3.2. Prediction of Phosphorylation Sites of Goat EGFR Protein 

The full-length cDNA of goat *EGFR* gene encodes 1208 amino acids. We predicted, using NetPhos 3.1 Server software, that the phosphorylation sites of EGFR protein are mainly located between amino acid residues 1000–1200, enriching tyrosine and serine sites (Figure 2). 

### 3.3. Effects of EGFR Gene on the Lipogenic Genes in GMECs

In GMECs transfected with Ad-EGFR recombinant adenovirus vector, the mRNA level of EGFR gene was increased significantly (*p* < 0.05, Figure 3A), and a significant increase in EGFR and phosphorylated EGFR proteins was also seen (*p* < 0.05, Figure 3B). In Ad-EGFR-transfected GMECs, the mRNA levels of *LXRa*, *LXRb*, *SREBP1*, *SP1*, *ACSS1*, *ACC*, *FASN*, *SCD1*, *DGAT1* and *DGAT2* genes were significantly increased (*p* < 0.05, Figure 3C–F), and the mRNA level of the *FABP3* gene was significantly decreased (*p* < 0.05, Figure 3D), while the mRNA levels of *PPARG*, *ACSS2*, *ACSL1*, *FADS1* and *FADS2* genes were unaffected (Figure 3C–E). 

To certify whether the expressions of the abovementioned altered genes were affected in absence of EGFR gene, three specific siRNAs, siRNA-1159, siRNA-243 and siRNA-2408, were designed to knockdown *EGFR* gene of GMECs. The results demonstrated that the mRNA level of *EGFR* gene in siRNA-transfected GMECs was significantly decreased, especially in GMECs transfected with siRNA-1159 (*p* < 0.05, Figure 4A). Next, the investigation of gene expressions in siRNA-1159-transfected GMECs demonstrated that the mRNA levels of *LXRa*, *LXRb*, *SP1*, *ACC*, *FASN* and *SCD1* genes were significantly decreased (*p* < 0.05, Figure 4B,D), and the mRNA level of *DGAT2* gene was significantly increased (*p* < 0.05, Figure 4E), but the mRNA levels of *SREBP1*, *ACSS1*, *FABP3* and *DGAT1* genes were unaffected (Figure 4B,C,E). 

### 3.4. Effects of EGF or Inhibitor on the Signal Protein Activity in GMECs

After GMECs were treated with 50 ng/mL EGF for 60 min, we found that the phosphorylation levels of ERK1/2, Akt and PLC-γ1were increased compared to EGF-untreated GMECs (*p* < 0.05, Figure 5A–C). Nevertheless, when GMECs were firstly treated with the specific inhibitor MK2206, U0126 or U73122, and then stimulated with 50 ng/mL EGF for 60 min, the phosphorylation levels of ERK1/2, Akt and PLC-γ1 were decreased compared to EGF-treated GMECs (*p* < 0.05, Figure 5A–C). The abovementioned results demonstrated that the inhibitors MK2206, U0126 or U73122 are capable of blocking the protein activity of the EGFR-mediated signaling pathway in GMECs.

### 3.5. Expressions of Lipogenic Genes and TG Content in Inhibitor-Treated GMECs 

In GMECs co-treated with 500 nM MK2206 and 50 ng/mL EGF for 24 h, we found that the mRNA levels of *FASN* and *SCD1* genes were significantly decreased compared to EGF-treated GMECs (*p* < 0.05, Figure 6A), and the mRNA levels of *ACC*, *SP1*, *LXRa* and *LXRb* genes were unaffected (Figure 6A). In GMECs co-treated with 10 µM U0126 and 50 ng/mL EGF for 24 h, the mRNA level of *SCD1* gene was significantly decreased compared to EGF-treated GMECs (*p* < 0.05, Figure 6B), and the mRNA levels of *FASN*, *ACC*, *SP1*, *LXRa* and *LXRb* genes were unaffected (Figure 6B). In GMECs co-treated with 20 µM U73122 and 50 ng/mL EGF for 24 h, the mRNA levels of *FASN* and *SCD1* genes were significantly decreased compared to EGF-treated GMECs (*p* < 0.05, Figure 6C), and the mRNA level of *SP1* gene was significantly increased (*p* < 0.05, Figure 6C), while the mRNA levels of *ACC*, *LXRa* and *LXRb* genes were unaffected (Figure 6C). 

Then, in GMECs co-treated with MK2206, U0126 and EGF, we found that the mRNA levels of *FASN* and *SCD1* genes were significantly decreased compared to EGF-treated GMECs (*p* < 0.05, Figure 7A), and the same result was also obtained in GMECs co-treated with U73122, U0126 and EGF (*p* < 0.05, Figure 7B). In GMECs co-treated with U73122, MK2206 and EGF, the mRNA levels of *FASN*, *SCD1* and *ACC* genes were significantly decreased compared to EGF-treated GMECs (*p* < 0.05, Figure 7C), and the same result also was certified in GMECs co-treated with MK2206, U0126, U73122 and EGF (*p* < 0.05, Figure 7D).

Furthermore, we investigated TG level in GMECs co-treated with U73122, MK2206 and EGF. The results demonstrated that TG content was significantly decreased compared to EGF-treated GMECs (*p* < 0.05, Figure 7E). 

## 4. Discussion

### 4.1. EGF and Lipid Metabolism

Previous studies of mammals have revealed the importance of EGFR signaling for the regulation of cell proliferation, differentiation and invasion [24,25]. ERBB receptor tyrosine kinases and their ligands have important roles in normal development and in human cancer [26]. Knockout of EGFR gene demonstrated that EGFR played an important role during epithelial cell development in many organs [27,28].

Mammary gland growth and involution are based on a dynamic equilibrium between proliferation and apoptosis of MECs [29]. Previous studies demonstrated that EGF induces cell proliferation and cell cycle in numerous cell lines, and it can improve the cell survival and prevent autophagy by activating the Akt/mTOR signaling pathway [30,31]. The fatty acid synthesis is important for mammary gland development and functions during the lactation [32]. EGF is capable of modulating lipid production or secretion in cell lines derived from the intestine and the liver [11]. Binding of EGF and EGFR is known to induce transphosphorylation inside EGFR dimers [5]. The EGFR signaling by growth factor is essential for normal biological processes, and this mechanism modulates the strength and duration of EGFR signaling [33]. However, the mechanism of EGF affecting lipid in MECs remains unclear. In the present study, we demonstrated that the mRNA level of *EGFR* gene was increased during the lactation of the dairy goat. Involution is a step-wise process, characterized by intensive apoptosis of MECs [34]. The reduced *EGFR* mRNA level in dry lactation might be associated with the apoptosis of GMECs. The TG content in EGF-treated GMECs was significantly increased. Furthermore, in the presence of EGF and AG1478, the changes of lipid droplets in GMECs were demonstrated under a fluorescent microscope. These findings indicated that the EGFR gene is involved in the lipid in GMECs. 

### 4.2. EGFR Activation in Regulating Lipid Metabolism in GMECs 

It is known that EGFR is an active protein with tyrosine and serine sites, which was also demonstrated in our study. Penrose et al. have demonstrated that EGF increases the density of lipid droplets, which depends on EGFR expression and activation, as well as the individual cellular capacity for lipid synthesis in human colon cancer cells [35]. Guo et al. [36] found that EGFR activation promotes the fatty acid synthesis via SREBP1-mediated lipogenesis in glioblastoma. In our study, the roles of EGFR in regulating lipid metabolisms in GMECs were demonstrated by both adenovirus-mediated overexpression and siRNA-mediated interference. The analysis of mRNA abundance of *FASN*, *ACC* and *SCD1* genes in GMECs revealed that they were significantly upregulated by EGFR overexpression. In contrast, EGFR knockdown decreased the expression of these lipogenic genes in GMECs. Meanwhile, it was found that EGFR protein was significantly phosphorylated in EGFR-overexpressed GMECs. Our data suggested that the expressions of *FASN*, *ACC* and *SCD1* genes in GMECs are related to EGFR activation. 

### 4.3. PLC-γ1 and PI3K/Akt Signaling Is Involved in the Lipid Metabolism in GMECs

Upon binding to EGFR, EGF activates PI3K/Akt, PLC-γ1 and MEK/ERK1/2 pathways [37,38,39]. Previous studies have shown that inhibition of the Akt/mTOR pathway in GMECs decreases the expressions of genes related to fatty acid synthesis, such as *FASN*, *ACC* and *SCD1* genes [40,41]. In our study, suppression of Akt activity by its specific inhibitor MK2206 resulted in the downregulation of mRNA levels of *FASN* and *SCD1* genes in GMECs. U73122, an inhibitor of PLC-γ1, specifically abolishes PLC-γ1 phosphorylation [42]. In U73122-treated GMECs, the mRNA levels of *FASN* and *SCD1* genes were significantly decreased, even though EGF was added into the GMECs, which showed that the effect of PLC-γ1 phosphorylation on the expressions of *FASN* and *SCD1* genes is very evident. EGF activates the ERK1/2 pathway via Ras-mediated activation [43,44]. Moreover, the inhibition of ERK1/2 protein by U0126 only downregulated the expression of *FASN* gene. These results suggest that the effects of PI3K/Akt and PLC-γ1pathway on genes related with fatty acid synthesis are stronger than that of the ERK1/2 pathway.

Consequently, to further investigate the effect of the abovementioned three pathways on genes related to fatty acid synthesis in GMECs, GMECs were co-treated with two or more inhibitors. The results demonstrated that co-treatment of U73122 and MK2206 in GMECs resulted in a synergistic effect on inhibition of mRNA expression of *FASN*, *ACC* and *SCD1* genes, especially on *ACC* genes. However, co-treatment of U73122 and U0126 or MK2206 and U0126 in GMECs only affected the mRNA expression of *FASN* and *SCD1* genes, but did not affect mRNA expression of *ACC* genes; similar results were observed in U73122 or MK2206-treated GMECs. At the same time, it was found that TG content was decreased in GMECs co-treated with U73122 andMK2206. Wang et al. [45] reported that EGFR directly activated PLC-γ1 by phosphorylating its tyrosine residues and indirectly activated Akt by phosphorylating PI3K. PLC-γ1 and PI3K are both involved in the regulation of lipid metabolism and use the same substrate, phosphatidylinositol (4,5) bisphosphate (PIP2) [46], suggesting the importance of PLC-γ1 and PI3K/Akt pathways on lipid metabolism. Therefore, we hypothesize that the activation of PLC-γ1 and PI3K/Akt pathways play the main roles in lipid metabolism in GMECs. 

## 5. Conclusions 

In conclusion, EGFR is expressed with fold changes ≥30 in lactation compared to dry period and promotes TG synthesis during lactation period. The overexpression of EGFR gene increased the mRNA levels of *SCD1*, *FASN* and *ACC* genes, and siRNA knockdown of *EGFR* gene showed the opposite effects. EGF activates EGFR signaling, which phosphorylates PLC-γ1 and Akt, induces the expression of lipogenesis-related genes and increases intracellular TG content. Therefore, the present study provided strong evidence that EGFR is involved in fatty acid synthesis of GMECs, the molecular events of which are regulated mainly through the Akt- and PLC-γ1-signaling pathways.

## Figures and Tables

**Figure 1 animals-10-00930-f001:**
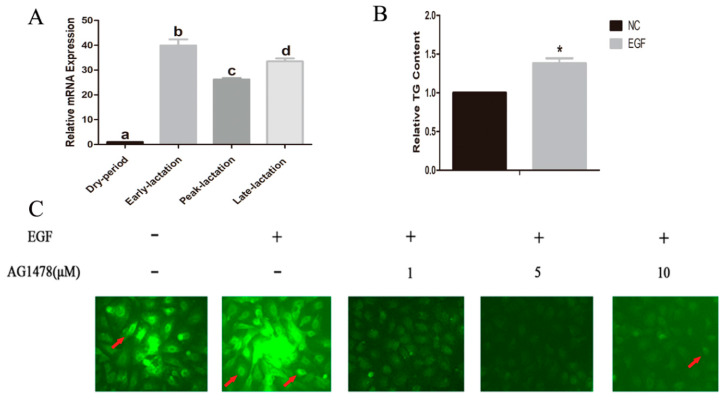
The mRNA expression of EGFR gene in goat mammary gland and effect of EGF on triglyceride (TG) and lipid droplets in goat mammary epithelial cell. (**A**) Goat mammary tissue samples at dry period, early lactation, peak lactation and late lactation were used to investigate the mRNA expression of EGFR gene. It is indicated that the values indicate the means of six individual samples. Different lowercase letters between bars represent *p* < 0.05. (**B**) GMECs were stimulated with 50 ng/mL EGF for 36 h, to detect TG content in GMECs. Cells cultured in FBS-free medium with no EGF as a negative control. TG was assayed by using a TG assay kit. The values indicate the mean ± SD of three individual samples; * means *p* < 0.05. (C) Lipid droplet formation of GMECs co-treated with AG1478 and EGF was measured by using BODIPY^®^ 493/503 assay. Scale, 50 microns. Red arrows indicate lipid droplets.

**Figure 2 animals-10-00930-f002:**
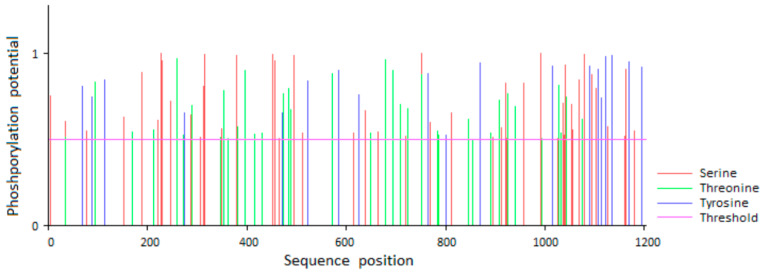
Prediction of the phosphorylation sites of goat EGFR. The phosphorylation sites of goat EGFR protein were predicted by NetPhos 2.0. Red color represents serine residues. Green represents threonine residues. Purple represents tyrosine residues.

**Figure 3 animals-10-00930-f003:**
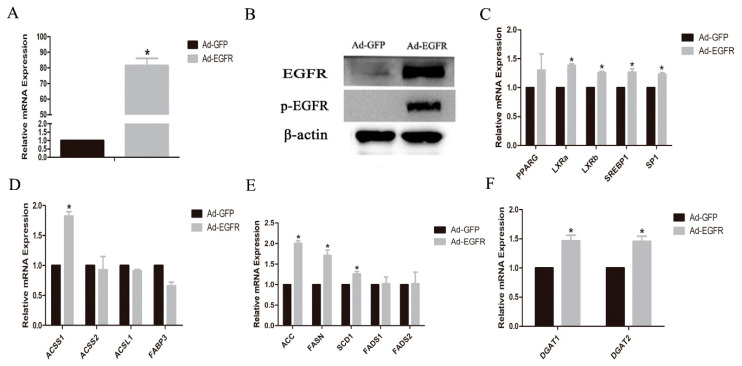
Effect of EGFR overexpression on genes related to lipid metabolism in GMECs. (**A**,**B**) EGFR mRNA and protein levels in EGFR-overexpressed GMECs were analyzed, using RT-PCR. Effect of EGFR overexpression on the mRNA expression levels of transcription factors (**C**), de novo fatty acid synthesis and desaturation (**E**), fatty acid uptake and activation (**D**) and TG synthesis (**F**). Ad-GFP is negative control. Values are presented as means ± SD of three independent experiments. * Means *p* < 0.05.

**Figure 4 animals-10-00930-f004:**
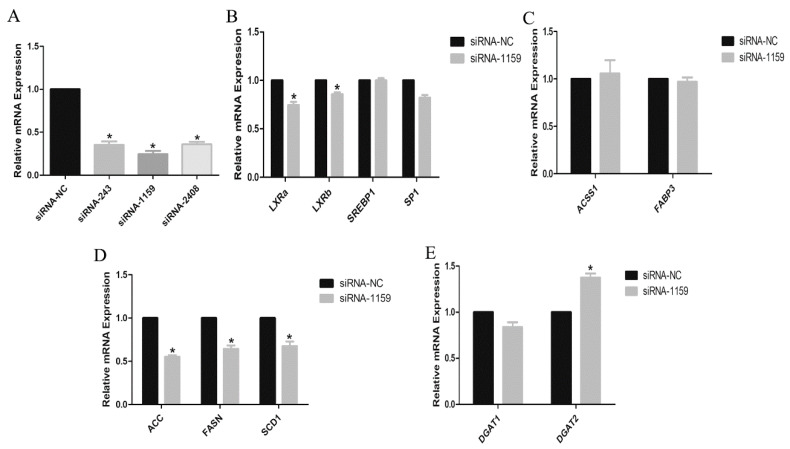
Effect of EGFR siRNA on genes related to lipid metabolism in GMECs. Control siRNA is negative control (siRNA-NC). (**A**) EGFR knockdown efficiency by siRNA was analyzed. (**B**–**E**) the expressions of lipogenic genes in EGFR knocked-down GMECs were investigated. Values are presented as means ± SD of three independent experiments. * Means *p* < 0.05.

**Figure 5 animals-10-00930-f005:**
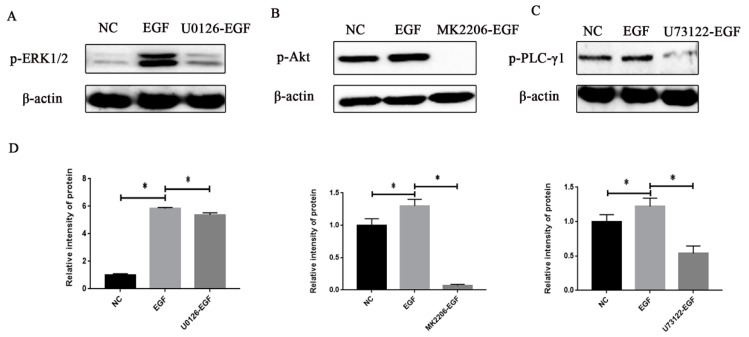
Effect of the protein inhibitor on signal protein activities in GMECs. (**A**) GMECs were treated with 500 nM MK2206 (Akt inhibitor) for 60 min and then stimulated with 50 ng/mL EGF for 60 min, to investigate the phosphorylation level of Akt. (**B**) GMECs were treated with 10 µM U0126 (MEK inhibitor) for 60 min and then stimulated with 50 ng/mL EGF for 60 min, to investigate the phosphorylation level of ERK1/2. (**C**) GMECs were treated with 20 µM U73122 (PLC-γ1 inhibitor) for 15 min and then stimulated with 50 ng/mL EGF for 60 min, to investigate the phosphorylation level of PLC-γ1. (**D**) The band intensity was shown by Image J software. Values are presented as means ± SD of three independent experiments. * Means *p* < 0.05.

**Figure 6 animals-10-00930-f006:**
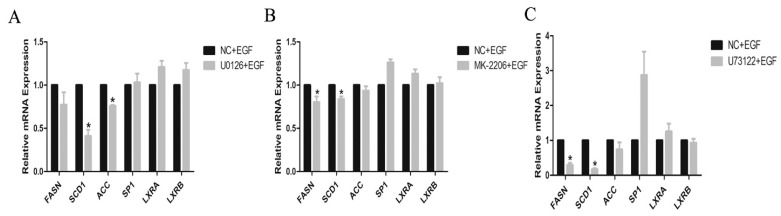
Effect of the protein inhibitor on the expressions of genes involved in lipid metabolism in GMECs. (**A**) GMECs were treated with 500 nM MK-2206 (Akt inhibitor) and 50 ng/mL EGF for 24 h, to investigate the mRNA levels of genes involved in lipid metabolism in GMECs. (**B**) GMECs were treated with 10 µM U0126 (MEK inhibitor) and 50 ng/mL EGF for 24 h, to investigate the mRNA levels of genes involved in lipid metabolism in GMECs. (**C**) GMECs were treated with 20 µM U73122 (PLC-γ1 inhibitor) and 50 ng/mL EGF for 24 h, to investigate the mRNA levels of genes involved in lipid metabolism in GMECs. Values are presented as mean ± SD of three independent experiments. * Means *p* < 0.05.

**Figure 7 animals-10-00930-f007:**
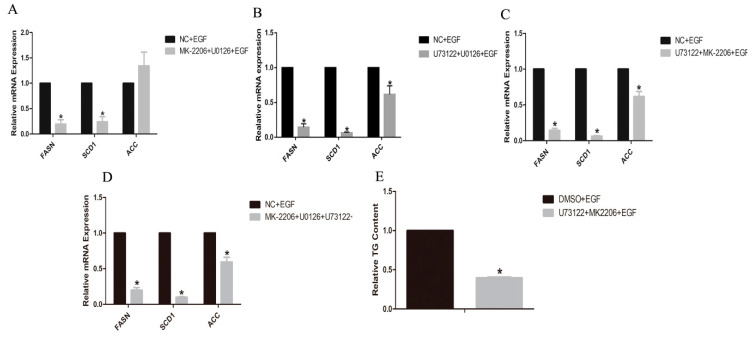
Effects of the combination of two or more inhibitors on the expressions of genes involved in lipid metabolism in GMECs. (**A**) GMECs were treated with 500 nM MK-2206 (Akt inhibitor), 10 µM U0126 (MEK inhibitor) and 50 ng/mL EGF for 24 h, to investigate the mRNA levels of genes involved in lipid metabolism in GMECs. (**B**) GMECs were treated with 20 µM U73122 (PLC-γ1 inhibitor), 10 µM U0126 (MEK inhibitor) and 50 ng/mL EGF for 24 h, to investigate the mRNA levels of genes involved in lipid metabolism in GMECs. (**C**) GMECs were treated with 20 µM U73122 (PLC-γ1 inhibitor), 500 nM MK-2206 (Akt inhibitor) and 50 ng/mL EGF for 24 h, to investigate the mRNA levels of genes involved in lipid metabolism in GMECs. (**D**) GMECs were treated with 20 µM U73122 (PLC-γ1 inhibitor), 500 nM MK-2206 (Akt inhibitor), 10 µM U0126 (MEK inhibitor) and 50 ng/mL EGF for 24 h, to investigate the mRNA levels of genes involved in lipid metabolism in GMECs. (**E**) GMECs were treated with 20 µM U73122, 500 nM MK-2206 and 50 ng/mL EGF for 24 h, to investigate TG content in GMECs. Values are presented as mean ± SD of three independent experiments. * Means *p* < 0.05.

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
