# Peer review of "Epidermal Growth Factor Stimulates Fatty Acid Synthesis Mainly via PLC-γ1/Akt Signaling Pathway in Dairy Goat Mammary Epithelial Cells"

_animals, 2020, doi:10.3390/ani10060930_

Round 1
Reviewer 1 Report
The present manuscript investigates the functionality of EGF receptor on lipid metabolism in goat mammary epithelial cells. --The manuscript needs work on the rationale, hypothesis and objectives in the introduction. Why is this research important for the understanding of goat mammary physiology and goat milk production? Do goats produce more milk fat? Are fatty acids in goat milk different from other species? -- M&M needs work developing and giving more detail in every subsection. This conditions results and discussion.
ABSTRACT:
Please, make sure that the abstract is comprehensive by itself. There are terms (FASN, ACC, SCD1, LXRa, LXRb, SP1, U73122, MK-2206) that need to be introduced in material portion of abstracts instead of naming them directly in the results portion.
Line 30: are significantly up-regulated compare with what?
Line 31: remove “obviously”, it is a subjective interpretation.
Line 33: Please, introduce FASN, ACC, SCD1, LXRa, LXRb and SP1 when you explain that lipid-related genes were measured, They come up in the result portion without introduction and may not know what they are.
Line 35: ‘special inhibitors’ does not mean anything to the reader. Please, modify to specific EGFR inhibitors and in parenthesis add the names of the drugs.
Line 36-38: EGF treatment is pointed twice, at the beginning and end of the sentence.
Line 38: ‘Obviously’ is a subjective term.
Line 39: Italicize ‘de novo’.
INTRODUCTION
The first paragraph tries to introduce this work, however, it is very general information about goat production and milk synthesis. I believe it does not add anything to the rationale and the justification of why this works is needed.
Introduction does not show a strong rationale of why the effect of EGFR on lipid metabolism is so important in goat mammary tissue. Maybe including the differences in fatty acid composition in
Introduction is missing the statement of the hypothesis. This is completely fundamental. Moreover, it needs to have a clear statement the objectives. Specific objectives add the
MATERIAL AND METHODS:
This section needs to be more detailed. I am concerned about the accuracy of the results based on how undetailed the M & M section is.
Line 84: Were those tissues collected by biopsy or after euthanasia? Please, specify. What time of the day were the samples collected? Lactating samples, were collected after milking?
Please, specify the time relative to milking. After milking there is a release of multiple hormones (e.g. prolactin, glucocorticoids) that influence on mammary gland physiology.
Line 99: Please, include the origin of the GMEC. Are those primary cells of stablished cell lines?
In case of primary cell culture, please, add the method of extraction and isolation, as well as physiological state (pregnant vs. lactating) of the animal/s used.
Line 100: Why did you decide not to culture the cells in the presence of prolactin? Your results in lactating vs. dry-off goats show overexpression of EGFR in lactation, thus if you want to model this in cell cultures, the culture medium should include PRL.
Line 102-105: Did you select monoclonal lines after transfection? Add the passage transfection and experiments were carried out
Lines 106-119: please, change “special inhibitors” for “specific inhibitors” and add after each inhibitor, which molecule they inhibit. I am confused if all those inhibitors are specific for the EGFR or the intermediaries. This really makes the manuscript difficult to follow.
Line 121: Please add the amount of RNA that was used to synthetize cDNA and the reagents used.
Line 123: please, add the SD of the Ct for the housekeeping gene.
Line 126: Add how q-PCR data (Ct) was analyzed. Which method was used?
Line 129: How was protein extracted and how much protein was used? How did you analyze density of western blot bands?
Lines 135-139: I think you can add to this section how the determination of lipid droplets and quantification was performed.
Statistical analysis: add the normalization of the data, which treatment was used as normalizer in each experiment? -- How was the relative expression of genes calculated?
Please, explain post-hoc analysis. What type of ANOVA was used?
RESULTS and FIGURES:
Line 148 and Figure 1A: dry-period samples were collected at 60d prior parturition, but when did the animals were dry off? – Dry period is very dynamic with involution/development happening, thus the time those samples were collected based on the start of dry off and time to parturition could determine the result.
Lines 151-152 and Figure 1C. It is hard to distinguish “lipid droplets” in those pictures, Do you have pictures with greater magnifications? – How was the analysis of the lipid droplets formation assessed?
Line 152: what is AG1478?
Figure 1: What is ‘NC’?
Figure 3: What Ad-GFP means? If a internal control or cells with mock sequence, please add it in the M&M.
Line 236: modify “certified”.
All Figures: please, add the number of replicates per ‘independent experiment’. What is considered as ‘independent experiment’? -- Were ‘independent experiment’ cultures plated/studied in a different time-period, or cells in different cell culture dish/well? – Because all the data is relative, I believe that control bars (solid black) bars can be remove. The value in all of them is 1 and does not add much.
DISCUSSION:
Discussion is vague and more focus on reviewing information about the topic and present the results. Discussion is meant to understand results based in previous authors.
Please, it is important to integrate differences in EGFR in mammary tissue with the development of the mammary gland.
Lines 278-279: Mammary gland samples include multiple type of cells (epithelial, myoepithelial, fibroblasts, etc). I believe you should compare your EGFR qPCR results with modifications in mammary tissue. For example. Dry-period concurs with involution and lost in epithelial cells, that could be the reason of lower expression.
Line 281: change ‘certified’.
‘PLC-γ1 and PI3K/Akt signaling are mainly involved into the lipid metabolism in GMECs’ Subsection: Here, finally, the targets of those inhibitors are described. It is difficult to follow all the previous sections without that explanation.
CONCLUSION:
Line 325: I do not think the model used with GMEC test for lactation. Cells were exposed to insulin and hydrocortisone, but no PRL was apply (based in M&M) section, so the lactogenic treatment was not correct and does not model lactation state.
TABLES
Table 1 and 2: from my point of view, both tables can me transfer as supplemental file. Unless this manuscript aims to develop primers and si-RNA sequences, both tables do not add much to the present manuscript story.
Author Response
Dear Reviewer:
Thank you very much for giving us an opportunity to revise our manuscript, we appreciate editor and reviewers very much for their positive and constructive comments and suggestion on our manuscript. I really appreciate all your comments and suggestions! The suggestions have enabled us to improve our work. Based on the instructions provided in your letter, we uploaded the file of the revised manuscript. Please find my itemized responses in below and my revisions/corrections in the re-submitted files.
Thanks again!
Yours sincerely
Jiangtao Huang
Corresponding author: Huaiping Shi
E-mail: huaipingshi@nwafu.edu.cn

Reviewer 2 Report
This study is focused on the analyses of the cell pathway involved in the function of EGF at regulating the Fatty Acid secretion and production in goat mammary epidermal cells.
The study is very interesting, with a nice and complete experimental design to answer the question proposed initially.
However some minor changes should be applied to improve the quality and understanding of the manuscrit.
Title: I would remove 'exogenous' because it seems that EGF has been infused in vivo. Introduction: Animals is not a journal of cell biology so a minimum explanation of the roles of proteins mentioned along the introduction should be included (i.e.: transcription factor xx, second messenger xxx,..) and also the abbreviations list. Methods: In the chemical section explain what is AG1478, MK2206, U0126 and 77 U73122 Why UXT is used as a housekeeping in RT_PCR? has it been compared with other HK genes? Why the authors used beta actin in Western blot and UXT in RT-PCR? Please, include in methods or in table 2 the protein /function codified by each gene to better understand the results. Figures: Some panels are deformed, letters not proportional. I suggest to include an scheme/draw of the conclusions found about the cell signalling, factors interactions and EGF effect. Finally some typos and grammar errors can be detected throughout the document. Please revise carefully the text. Some examples: Line 19 Remove 'for instance' Line 24 mediating instead mediated Line 26 expressions instead Expressions Line 29 expression instead expressions L45 content instead contents L56 extracellular signals or intracellular? L278. Lipid .......Author Response
Dear Reviewer:
Thank you very much for giving us an opportunity to revise our manuscript, we appreciate editor and reviewers very much for their positive and constructive comments and suggestion on our manuscript. I really appreciate all your comments and suggestions! The suggestions have enabled us to improve our work. Based on the instructions provided in your letter, we uploaded the file of the revised manuscript. Please find my itemized responses in below and my revisions/corrections in the re-submitted files.
Thanks again!
Yours sincerely
Jiangtao Huang
Corresponding author: Huaiping Shi
E-mail: huaipingshi@nwafu.edu.cn

Reviewer 3 Report
The authors demonstrated in their research study that lipid metabolism in GMECs during lactation is driven by activation of Akt and PLC-y1-pathways and showed, which genes are affected by those pathways.
The method section is lacking a huge amount of details that need to be provided. Please see my comments below.
Also, the discussion needs to be revised. You need to take the literature and information that you nicely show at the beginning of your sections and put it in the context to your results.
Unfortunately, there are heavy issues with the style and the English language. I commented on the language and suggested some changes below. This list is not comprehensive though. Please keep in mind that English is not my native language.
General:
- Corrct author?: Darryl Lynn Hadsell
- Keywords: EGF is used twice
Simple summary:
- L16: regulates
- L17/18: new sentence: This study aimed to
- L19: For instance, = Therfore,
- L19: triglyceride
- L20: Akt
Abstract:
- L23: finally involving in the regulation = and is involved in the regulation
- L25: have been poorly = are poorly
- L25: TG – please introduce the abbreviation ahead = triglyceride (TG)
- L26: using the TG assay = using TG assay
- L26: Besides = Further
- L26: expression
- Sentence L26-29: Please indicate which expression was measured with which method
- L31: “obviously” is not a scientific expression, please use “significantly”
- L31: and the increase = and an increase
- L32: please delete “in EGF-treated GMECs”
- L33: please write names of genes in cursive: FASN, ACC, SCD1, LXRa, LXRb and Please check your whole manuscript.
- L35: special = specific
- L37: please write the names of genes cursive
- L37-38: co-treated with U73122, MK-2206 and EGF
- L38: “obviously” is not a scientific expression, please use “significantly” and indicate p-value (otherwise just state that TG content was also decreased/low)
- L40: Please be consistent with writing: is it AKT signaling or Akt signaling or akt signaling? Please check your whole manuscript.
The abstract overall does not really summarize all the work that you put in. For example, you do not mention overexpression or silencing of EGFR.
Introduction:
- Sentence L46-47: Please revise this sentence, i.e. These fatty acids are synthesized in goat mammary epithelial cells. Then, TG droplets are formed and transported out of the cell, where the fatty acids become a component of goat milk. – Keep in mind that this is a suggestion and might not be ideal.
- L49: related to fatty acids (please indicate how they are related: fatty acid synthesis, fatty acid transport etc.)
- L49: please delete “into”
- L50: responsible for
- Side sentence L50-L51: , however, it is still not clear how they regulate those genes.
- Sentences L53-L54: I would suggest combining the 2 sentences with “and”
- L57: I would suggest to exchange “openness” with “activation”.
- L60: droplets
- L62: FASN
- L66: However,
- L68: involved
- L70: related to
Materials and methods
Section 2.1
Could you specify the names of the antibodies? Also, I suggest to not use the words “Antibodies specific for” all the time. Please use variations, i.e. “an antibody against EGF” or “antibody binding to” etc.
It would also be good to know, where these antibodies are binding exactly (maybe you could add supplementary files).
Section 2.2
The number are not clear to me. Please specify: How many samples were taken from how many herds? Is there a reason why you sampled 24 herds? What was the size distribution (animals per herd)?
You should also specify the procedure of collecting mammary gland tissue.
Section 2.3
Please provide the official number of the cDNA used and specify the database (NCBI, Ensembl…). Or did you amplify the cDNA sequence yourself? Isn’t there a (predicted) protein sequence from a database available?
Section 2.4
The abstract and summary do not include information about siRNA treatment. You should add this information.
L95: Please add a website, where those algorithms can be found.
L95: sequences for interference are listed
Section 2.5 and section 2.6
I cannot really track your method here. Please be more specific about what you cultured for how many hours and when you harvested your cells.
Please don’t mix all treatments together: Cells were treated with … for x minutes or with… for y minutes or otherwise… That makes it very difficult for the reader to follow. Please entangle your treatments so that the reader can follow.
What was your control? What was/were your treatment group(s)?
Section 2.7
L121: Please specify, from which tissue and cells you extracted mRNA.
Why did you choose UXT as housekeeping gene? Why did you use one housekeeping gene (instead of 2-3)?
Also, please explain why you used the genes (FASN, ACC, SP1...) that you used for analysis?
L125: Primers for real-time quantitative PCR are shown
Table 2: The layout of the table seems to be off, long gene names are divided and shown in two rows. Please adjust the layout so that the gene name is only in one row.
Section 2.8
Which protein samples? How did you isolate protein? From which cells/tissues did you isolate protein? How many samples did you use for Western blot analysis? Please add information. Which antibodies were used against which protein? What is the difference between EGFR and pY1068-EGFR?
Section 2.9
Sometimes you write P<0.05 and sometimes P > 0.05. Please be consistent.
Results:
Section 3.1
L147: EGFR
L147: in all lactation periods
L148: than the levels in tissue of the dry period
L148-149: Please rewrite sentence.
L150: compared to EGF-untreated GMECs
L151-L155: needs to be rewritten so that the reader can follow.
L163: *P > 0.05
L164: Please specify the treatment length. Are the numbers over the pictures corresponding to the amount of AG1478 added?
Section 3.2
L167: encodes 1298 amino acids OR encodes a polypeptide with 1208 amino acids OR encodes a 1208 amino acids long polypeptide
Sentence L167-L169: Using Phosnet 3.0 it was predicted that the phosphorylation… located between …. and are enriched for…
L173-L174: This sentence does not make sense, please revise.
Section 3.3
L177: “dramatic” is not a scientifically appropriate word. Please describe the fold change or use “significantly”
L179-182: Please write the gene names in cursive
L197: *P > 0.05
L202; *P > 0.05
There is no section 3.4 – please renumber the result sections
Section 3.5
Figure 5: Is there a reason why there are 2 bands for p-ERK1/2 (NC, EGF, U0126-EGF)?
Section 3.6
L220 co-treated with
Please add information why you combined inhibitors in your study design (in Method section).
L248: *P > 0.05
L261: *P > 0.05, You can delete this in L259. It is enough to mention it once per Figure.
Discussion
L265: Please explain ERBB abbreviation before using only the abbreviation
L271: lines,
L278: lipid
L278: remains poorly understood OR remains unclear
L281: I suggest to use another word for “certified”, i.e. shown, demonstrated
L284: see L281
L285: EGFR protein enriches the tyrosine and serine sites – I don’t really see how you came to this conclusion. Please specify in your results. Also what statistics did you use to verify this statement?
L286: droplets
L290: I suggest to use another word for “clarified”, i.e. shown, demonstrated
L295: genes in GMECs are related to EGFR activation
Subheading: PLC-y1 and PI3K/Akt signaling are involved in lipid metabolism in GMECs
L298: please delete one “and” between the pathways
L304: even though
L305: I suggest to use “showed” instead of “suggested”
L306: I suggest to use “evident” instead of “very obvious”
L307: expression of
L308: I suggest to use “suggest” instead of “indicated possibly”
L310: I suggest to use “Consequently,” instead of “Next,”
L310: to further investigate
L310: related to
L313: ACC genes or ACC gene?
L315 but did not affect mRNA expression of ACC
L315-L316: please rewrite the last part of the sentence (, these results…)
L317: co-treated with
L317: please don’t use Figures in the Discussion
L318: Wang et al. reported
L320: Please don’t use abbreviations before explaining them (PIP2)
L321: on lipid metabolism
L321: it is speculated – this phrase os very weak for using it in a discussion, espescially if you have results that support your hypothesis. Maybe you can use: Therefore, we hypothesize that the activation…
L325: I don’t think abundant is the right word to use in that context. Basically, you show that EGFR is expressed during lactation, maybe you can add the average fold change (compared to dry period)
L325: Instead of “The overexpression” please use “Overexpression”
L327 EGFR gene showed opposite effects
L334: Darryl Lynn Hadsell
Author Response
Dear Reviewer:
Thank you very much for giving us an opportunity to revise our manuscript, we appreciate editor and reviewers very much for their positive and constructive comments and suggestion on our manuscript. I really appreciate all your comments and suggestions! The suggestions have enabled us to improve our work. Based on the instructions provided in your letter, we uploaded the file of the revised manuscript. Please find my itemized responses in below and my revisions/corrections in the re-submitted files.
Thanks again!
Yours sincerely
Jiangtao Huang
Corresponding author : Huaiping Shi
E-mail: huaipingshi@nwafu.edu.cn

Round 2
Reviewer 1 Report
The present revision of the manuscript has improved compare to the previous version.
The specific comments to the manuscript are the following:
SIMPLE SUMMARY: I had a hard time interpreting the message in this section. There is some sentences that are lacking sense and cohesion. For example:
“The Goat mammary epidermal cells (GMECs) play a crucial role in lactation, secreting growth factors that promote and feed back to mammary gland development” -- It is not necessary to say that GMECs play a crucial role in lactation, it is implicit in the role of MECs in general, and which type of feed-back you refer to? Positive or negaitive factors?
“Epidermal growth factor (EGF) positively regulates the development, production and secretion of fatty acid from the mammary gland” -- Fatty acids are not developed, synthesized, but not developed, cells or tissues are the ones that develop.
The simple summary dies not tell what this study found as results or the conclusions.
ABSTRACT:
Line 39: Please substitute “U73122, MK-2206” for “PLC-y1 and Akt inhibitor”
INTRODUCTION:
Line 60: reference Papaiahgari et al. is missing the year
The introduction still missing the statement of the hypothesis and rationale of how with known previous research you thought the objective was the adequate. The objectives need improvement. Your general objective is to study the role of EGFR in lipid metabolism in GMECs, and it was achieved by specific aims, look at the abundance of EGFR in lactating and non-lactating tissue; effect of EGF in lipogenic synthesis; and the use of inhibitors for EGFR pathway molecules to understand modifications in lipid metabolism. You need to state each one of the specific objectives, so the reader can follow the next sections.
M & M:
Line 89: thanks for adding the day when the dry period samples were taken, but it is needed that you add the days those animals were from start of dry off. Most dairy animals start their dry off at 60 days previous parturition, so what it is found in their tissues is active involution process. During involution, there is an active regression and loss in GMEC, so the decreased in EGFR could be related with that process. If samples were collected after the involution process was completed, that is a “real” dry off. Please, specify how many days from start of dry off samples were collected. The last portion of the dry period (last 3 weeks in most dairy species), is marked by increased epithelial cell proliferation and mammary development, and in this period we would expect increased expression of EGFR.
Line 96: Please, add where the EGFR cDNA sequence was obtained.
Line 114: If the porpoise was to create a lactation phenotype in the GMECs, please, explain why the FBS-free medium was not supplemented with glucocorticoids and insulin. Insulin is fundamental for the health of mammary epithelial cells, without it, they have difficulties to grow. Moreover, glucocorticoids (hydrocortisone or dexamethasone) are as well necessary to achieve the lactation phenotype in MEC. A lactogenic medium must count with the presence of Insulin, prolactin, and glucocorticoids.
RESULTS:
Line 230: Please, add after each inhibitor which molecule they block.
DISCUSSION:
“EGF and lipid metabolism” section needs improvement. There is a lot of information in review article format, but not much discussion with findings.
Lines 301-305: I believe that reduced EGFR in the dry off glands could be related with active involution process. AS commented before add the day relative to start of dry off that the animals were biopsied. If the goat were in active involution, this needs to be commented in discussion and set up as a limitation of the study.
How do you integrate this findings with the lactation cycle. You state that EFG and EGFR is associated with lipid synthesis. EGF is traditionally related with proliferative states (proliferation during pregnancy) in the mammary epithelial cells, and no as much with differentiation and secretory activation and lipid synthesis states (for milk synthesis). Lactation is represented for less proliferation and greater proliferation and milk component synthesis, compared with late pregnancy, so we would expect less expression of EGFR.
Figures:
Fig 1: Please add arrows that help locating the lipid droplets in section C. – Add the meaning of abbreviation NC in the figure legend.
Fig 3: Please, explain in the legend the use of ad-GFP as control
Fig 4: add to teh legend what siRNA-NC means
Fig 5, 6, 7: please add, after any of the inhibitors are mentioned, which molecule they inhibit. I believe will help with the understanding of figures, without the necessity to consult the main text.
Author Response
Dear reviewer,
We have studied the valuable comments from you. According to the comments from you, we have carefully corrected some wording, sentence structure and grammatical issues. Special thanks to you for your good comments.
Sincerely yours,
Jiangtao Huang
Corresponding author: Huaiping Shi
E-mail: huaipingshi@nwafu.edu.cn

Reviewer 3 Report
The authors were able to improve the flow and readability of the paper.
However, I have a couple of points and remarks:
- when I google dhadsell@bcm.edu I end up with Darryl L. Hadsell, not Darry: please check name of author again
- References are not in MDPI style, please change
- L11 in affiliations: College, not Colloge
- L18 I suggest to start a new sentence: …acid from the mammary gland. This study aimed to…
- L20 droplets
- L35 genes were positively correlated to the mRNA level of EGFR gene shown by gene overexpression and silencing.
- L50 related to, not related with
- L51-52 –acid synthase (FASN), and other enzymes responsible…
- L78 What does MEK stand for? Maybe you can put it to the abbreviation section.
- L93 as previously described
- L113-114 I don’t understand: GMECs were replaced with FBS-free medium; Did you mean: Six hours after transfection, GMECs were washed and FBS-free medium was added?
- L118 GMECs were grown
- L118 80% confluence
- L118-119 see comment L113-114
- L121 , were added respectively
- L125 co-treated with
- L125 for 36 h were collected
- L142 Please remove point before Primers for RT-PCR
- L147 methods of Zhang
- L149 total protein
- Results section
3.1 = 3.1, 3.2 = 3.2, 3.3 = 3.3, 3.5 = 3.4, 3.6 = 3.5
- L171 tissues in all lactation periods
- 172-174 Moreover, the expression level of EGFR was ~35-fold higher in early lactation and ~30-fold higher in late lactation compared to dry period (P < 0.05, Figure 1A).
- L175 I suggest to use Furthermore instead of Meanwhile
- L178-179 In fact, the total fluorescence intensity dropped steadily with the increase of AG1478 concentration (Figure 1C).
- L189 In L185 it is indicated that the values indicate the means of six individual samples. Also, for Figure 1C there are no values.
- L193 software that the phosphorylation
- L193 mainly located between amino acid residues 1000 – 1200
- L197 NetPhos 2.0 = Phosnet 3.0 ?
- L201 and a significant increase in
- L210 transfected with
- L230 with the specific inhibitor
- L233 that the inhibitors MK2206, U0126 and U73122 are capable of
- L244 co-treated with (also L247, L250, L254, L256, L257, L259)
- L262 you already mention “GMECs co-treated with U.., MK… and EGF” in the sentence before, so you do not need it here
- L294 “defend” autophagy, do you mean “prevent”, “induce”? Please change the verb.
- L301 1ipid = lipid
- L305 involved in the lipid
- L338 genes, but did not affect mRNA expression of ACC genes, similar results
- L340 co-treated with U73122 and MK2206
- L343 use the same substrate, phospho…
- L350 EGFR (cursive) gene showed opposite
If Tables are Supplementary they should be renamed to Supplementary Table 1 and Supplementary Table 2
If possible broaden Table 1 so that you need one line per row
Author Response
Dear reviewer,
We have studied the valuable comments from you. According to the comments from you, we have carefully corrected some wording, sentence structure and grammatical issues. We appreciate for your warm work earnestly, and hope that the correction will meet with approval. Once again, thank you very much for your comments and suggestions.
Sincerely yours,
Jiangtao Huang
Corresponding author: Huaiping Shi
E-mail: huaipingshi@nwafu.edu.cn
